# Prognostic Value of a Novel and Established High-Sensitivity Troponin I Assay in Patients Presenting with Suspected Myocardial Infarction

**DOI:** 10.3390/biom9090469

**Published:** 2019-09-09

**Authors:** Nils A. Sörensen, Sebastian Ludwig, Nataliya Makarova, Johannes T. Neumann, Jonas Lehmacher, Tau S. Hartikainen, Paul M. Haller, Till Keller, Stefan Blankenberg, Dirk Westermann, Tanja Zeller, Niklas Schofer

**Affiliations:** 1Department of General and Interventional Cardiology, University Heart Center, Martinistrasse 52, 20246 Hamburg, Germany; n.makarova@uke.de (N.M.); jo.neumann@uke.de (J.T.N.); j.lehmacher@uke.de (J.L.); t.hartikainen@uke.de (T.S.H.); p.haller@uke.de (P.M.H.); s.blankenberg@uke.de (S.B.); d.westermann@uke.de (D.W.); t.zeller@uke.de (T.Z.); n.schofer@uke.de (N.S.); 2German Center for Cardiovascular Research (DZHK), Partner Site Hamburg/Kiel/Lübeck, Martinistrasse 52, 20246 Hamburg, Germany; 3Kerckhoff Heart and Thorax Center, Department of Cardiology, Benekestrasse 2–8, 61231 Bad Nauheim, Germany; t.keller@kerckhoff-fgi.de; 4German Center for Cardiovascular Research (DZHK), Partner Site RheinMain, Benekestrasse 2–8, 61231 Bad Nauheim, Germany

**Keywords:** acute coronary syndrome, troponin, prognosis, risk assessment, cardiovascular events

## Abstract

High-sensitivity troponin has proven to be a promising biomarker for the prediction of future adverse cardiovascular events. We aimed to assess the prognostic value of high-sensitivity troponin I (hs-TnI) on admission in patients with suspected acute myocardial infarction (AMI) analyzed by a novel (Singulex Clarity cTnI) and established hs-TnI assay (ARCHITECT STAT hs-TnI, Abbott). Hs-TnI was measured in a total of 2332 patients from two prospective cohort studies presenting to the emergency department with suspected AMI. The prognostic impact for overall and cardiovascular mortality of both hs-TnI assays was assessed in the total patient cohort as well as in the subgroups of patients with AMI (*n* = 518) and without AMI (non-AMI) (*n* = 1814). Patients presenting with highest hs-TnI levels showed higher overall and cardiovascular mortality rates compared to those with lower troponin levels, irrespective of the assay used. Both hs-TnI assays indicated association with overall mortality according to adjusted hazard ratio (HR) among the entire study population (HR for Singulex assay: 1.16 (95% CI 1.08–1.24) and HR for Abbott assay: 1.17 (95% CI 1.09–1.25)). This finding was particularly pronounced in non-AMI patients, whereas no association between hs-TnI and overall mortality was found in AMI patients for either assay. In non-AMI patients, both assays equally improved risk prediction for cardiovascular mortality beyond conventional cardiovascular risk factors. Hs-TnI is independently predictive for adverse outcomes in patients with suspected AMI, especially in the subset of patients without confirmed AMI. There was no difference between the established and the novel assay in the prediction of mortality.

## 1. Introduction

Cardiac troponin (cTn) is a biomarker released by injured myocardial cells and, thus, can be used to detect myocardial necrosis [1]. Therefore, measurement of circulating cTn represents a key diagnostic tool in patients with suspected acute myocardial infarction (AMI) [2]. For the diagnosis of AMI, international guidelines recommend using high-sensitivity troponin I (hs-TnI) or T (hs-TnT) assays allowing for a fast and accurate detection of AMI by applying effective rule-in and rule-out algorithms [3,4,5,6].

Moreover, high-sensitivity assays are capable of determining troponin levels far below the 99th percentile and some assays report a quantifiable result in over 95% of the general population. Hence, high-sensitivity assayed troponin, even at these lower concentrations, has also been proven to be a reliant biomarker in the prediction of adverse cardiovascular events in the general population [7,8,9,10] as well as in patients with suspected myocardial infarction [11,12,13]. The ability to predict mortality has been demonstrated for the assessment of both troponin subunits, cTnT and cTnI [9,14,15,16].

Recently, a novel hs-TnI assay (Singulex Clarity cTnI) has been developed promising accurate detection of even lower troponin I levels of <0.1 ng/L [17,18]. This hs-TnI assay enables troponin quantification in more than 99% of the general population [19] and, thus, holds potential for even better identification of patients at risk for future cardiovascular events.

However, data about the prognostic impact of this newly developed hs-TnI assay in comparison to well established high-sensitivity assays is scarce. The aim of the current study is to assess the potential role of a single hs-TnI measurement on admission by using the novel hs-TnI and an established hs-TnI assay (ARCHITECT STAT hs-TnI, Abbott) in cardiovascular risk prediction of patients with suspected AMI.

## 2. Materials and Methods

### 2.1. Study Population

A total of 2332 patients with available hs-TnI levels on admission, assessed by both the Singulex as well as the Abbott immunoassay, were included in our analysis. Our study encompassed patients of two study populations: the Biomarkers in Acute Cardiac Care (BACC) study and the stenoCardia study.

#### 2.1.1. Biomarkers in Acute Cardiac Care Study Population

The BACC study has been described previously [5]. A total of 1569 patients with suspected ACS presenting to the emergency department of the University-Hospital Hamburg–Eppendorf were included in the present study. All patients were enrolled between July 2013 and December 2016. Inclusion criteria were suspected ACS, age > 18 years, and the ability to provide written informed consent. The BACC study was registered at https://www.clinicaltrials.gov (unique identifier: NCT02355457).

#### 2.1.2. StenoCardia Study Population

Methods, follow-up, and adjudication of outcomes of the stenoCardia study have previously been reported in detail [20]. We included 763 patients presenting with acute chest pain to three German emergency departments (Mainz, Koblenz, and Hamburg) between January 2007 and December 2008. Inclusion criteria were the ability to provide written informed consent, age > 18 and < 85 years, chest discomfort within the last 24 h and suspected ACS, and the experience of at least 30 min of chest discomfort or other symptoms consistent with possible ACS. The stenoCardia study was registered at https://www.clinicaltrials.gov (unique identifier: NCT03227159).

Both cohorts correspond to the Declaration of Helsinki and were approved by the local Ethics Committees. All patients gave written informed consent and participated voluntarily.

### 2.2. Standard Diagnostic Approach and Adjudication of the Final Diagnosis

Patients presenting with suspected AMI were evaluated according to the current European Society of Cardiology (ESC) guidelines in both studies, including electrocardiography (ECG), serial cardiac troponin measurements, echocardiography, and, if indicated, additional instrumental diagnostic [4]. The final diagnosis was later adjudicated by two independent cardiologists in both studies. In case of disagreement, a third cardiologist was consulted. The diagnosis was validated according to the Third Universal Definition of MI [21].

In-house troponins for the diagnosis of AMI differed between the BACC and the stenoCardia study. In BACC, hs-TnT (Roche Elecsys, Roche Diagnostics, Germany) was used for adjudication, whereas in stenoCardia two different assays were used: fourth generation troponin T (Roche Elecsys, Roche Diagnostics, Germany) was used in Mainz and Hamburg and the ARCHITECT STAT cTnI assay (Abbott Diagnostics) was used in Koblenz.

### 2.3. Patient Follow-Up

In both studies, follow-up was obtained from telephone interviews by trained medical staff, questionnaires mailed to the patients, general practitioners, and/or electronic medical records. Follow-up included information about overall mortality as well as cardiovascular mortality. Ultimately, the local registry offices were consulted to assess mortality and acquire mortality certificates. Median follow up (interquartile range) time was 880 (864, 902) days.

### 2.4. Troponin I Measurements

Troponin I was measured in stored blood samples that were collected immediately after presentation to the emergency department, using two different hs-TnI immunoassays: the Singulex Clarity cTnI assay and the ARCHITECT STAT hs-TnI assay (Abbott). Samples were stored at −80 °C under standardized conditions until further analysis. Specific characteristics of both assays are described briefly in the following.

### 2.5. Singulex Clarity cTnI 

The manufacturer reports a test specific limit of detection (LoD) of 0.08 ng/L and a 10% coefficient of variation (CV) at a concentration of 0.53 ng/L. The 99th percentile has been described at 8.67 ng/L [17].

### 2.6. Abbott ARCHITECT STAT hs-TnI 

The test specific LoD has been described at 1.9 ng/L, with a 10% coefficient of variation (CV) at a concentration of 5.2 ng/L. In the general population the 99th percentile of this assay has been reported at 27 ng/L [22].

### 2.7. Statistical Analyses

For the initial description of baseline characteristics, we used quartiles for continuous variables and absolute and relative frequencies for binary variables. Survival curves for overall and cardiovascular mortality were computed for thirds according to hs-TnI levels using both investigational hs-TnI assays for the whole cohort as well as AMI and non-AMI subgroups using the Kaplan–Meier method. The log-rank test was used to compare survival curves. The median follow-up time was estimated by the reverse Kaplan–Meier method.

Comparison of both hs-TnI results was performed using the Passing–Bablock regression and by direct comparison using the Bland–Altman analysis.

Cox proportional hazards models for overall and cardiovascular mortality were computed for the entire cohort and for subgroups AMI and non-AMI using the available patients’ data. For these analyses troponin I was log-transformed. The Cox models for both considered endpoints were adjusted for age, sex, and conventional cardiovascular risk factors: body mass index, diabetes, smoking status, family history of coronary artery disease, hypertension, and hyperlipidemia.

The C-index was used to quantify the added predictive value of troponin I beyond that from a model including conventional cardiovascular risk factors. For these analyses, the two-year event probabilities were computed. For the computation of C-indices the follow-up times were censored at two years. Ten-fold cross validation was used to control for the over-optimism of calculating performance measures on the same dataset from which the models were computed. Differences in C-statistics (with 95% CIs) after the addition of troponin I to the model consisting of age, sex, and conventional cardiovascular risk factors were computed using the method described by Antolini et al.; C-statistics were provided for log-transformed troponin I values and the respective assay in the entire study cohort [23].

A two-sided *p*-value of <0.05 was considered statistically significant. All statistical methods were implemented in R statistical software version 3.5.1 (The R Foundation, Vienna, Austria) [24].

## 3. Results

### 3.1. Baseline Characteristics

Baseline characteristics of the study population are given in Table 1. Out of the 2332 included patients, 518 were diagnosed with AMI. Patients with AMI compared to those without AMI were older and more often male. Conventional cardiovascular risk factors were more frequently present in AMI patients. Median troponin levels analyzed by the Singulex assay were 2.3 ng/L among the whole cohort, as well as 88.3 ng/L and 1.6 ng/L, in AMI and non-AMI patients, respectively. Median hs-TnI levels assessed by the Abbott assay were higher throughout all groups (6.8 ng/L among all patients, 182.8 ng/L in AMI, and 5.0 ng/L in non-AMI patients).

Dividing all patients in thirds according to their hs-TnI level measured by both assays showed similar patient characteristics between both assays for patients in the lowest, middle, and upper hs-TnI third. However, median troponin levels in each third were higher with the Abbott assay (Appendix A). Patient characteristics according to thirds for the subgroups AMI and non-AMI patients are given in Appendix A.

A detailed comparison between both hs-TnI assays using regression analysis is given in the Online Appendix A.

### 3.2. Survival Analysis According to Troponin Thirds

Patients in the lowest third of troponin levels, irrespective of the hs-TnI assay used, had significantly lower overall mortality rates than the middle and the upper third (Figure 1A,B). When only cardiovascular mortality was considered, there was a significant difference between the lowest and the middle third according to the Singulex assay (*p* < 0.001). This was not observed with the Abbott assay. For both assays, patients in the upper third suffered significantly more often cardiovascular deaths (Figure 1C,D).

In patients with an adjudicated diagnosis of AMI, there was no difference in mortality in dependence of troponin levels (Appendix A). In the subgroup of patients without AMI, higher troponin levels were associated with poor overall survival and cardiovascular mortality (Appendix A).

### 3.3. Prognostic Value of hs-TnI

Cox regression analysis revealed strong association of hs-TnI levels with poorer outcomes. After adjustment for age, sex, and cardiovascular risk factors, hs-TnI proved to be an independent risk factor for overall mortality in the whole cohort with and hazard ratio (HR) of 1.16, (95% confidence interval (CI) 1.08–1.24, *p* < 0.001) for the Singulex assay and HR of 1.17 (95% CI 1.09–1.25, *p* < 0.001) for the Abbott assay. In non-AMI patients, HR was 1.38 (95% CI 1.19–1.61, *p* < 0.001) for the Singulex assay and 1.44 (95% CI 1.23–1.70, *p* < 0.001) for the Abbott assay. In contrast, in AMI patients HR neither reached significance for overall mortality for the Singulex assay with HR of 1.07 (95% CI 0.95–1.20, *p* = 0.28), nor for the Abbott assay with HR of 1.10 (95% CI 0.98–1.23, *p* = 0.11). For cardiovascular mortality HRs were significant for the whole cohort and both subgroups with HR of 1.26 (95% CI 1.14–1.38, *p* < 0.001) for the Singulex assay and 1.27 (95% CI 1.16–1.40, *p* < 0.001) for the Abbott assay (Figure 2).

## 4. Prediction of Adverse Outcome 

Assessment of the predictive value of hs-TnI in addition to a risk model including age, sex, and classical cardiovascular risk factors resulted in an improved prediction of fatal events for both assays. For overall mortality, we observed a tendency in C-index improvement, which failed statistical significance with a C-index difference of 0.01 (*p* = 0.08) for the Singulex and 0.012 (*p* = 0.074) for the Abbott assay. However, for cardiovascular mortality, C-index improvement was highly significant with a C-index difference of 0.038 (*p* = 0.001) for the Singulex and 0.042 (*p* < 0.001) for the Abbott assay) (Table 2).

## 5. Discussion

The present study assessed the association of troponin I levels, measured by two different high-sensitivity immunoassays, with adverse outcomes in patients with suspected AMI. We found that higher troponin I levels were associated with increased risk of both overall and cardiovascular mortality after 2 years of follow-up irrespective of the assay used. In particular, in the subgroup of patients without AMI, troponin I was independently predictive for adverse events, whereas in patients with confirmed AMI the association between troponin I levels and outcome was less pronounced. Both assays were found to be equally capable of predicting adverse outcome.

There is substantial interest in the inclusion of biomarkers into algorithms for the prediction of fatal cardiovascular events [25]. Troponin has emerged as a reliable biomarker for the identification of patients at risk for adverse outcomes in subjects with suspected AMI [26] and in the general population [8,27]. In two large meta-analyses comprising 154,052 and 74,738 subjects, respectively, Willeit et al. [28] and Blankenberg et al. [9] found associations of troponin levels with increased risk for incident cardiovascular disease and mortality. In line with these studies, our findings reinforce the value of troponin as a strong predictor for adverse outcomes, independent from common cardiovascular risk factors. Both troponin I assays investigated in this study showed similar hazard ratios in terms of prediction of adverse outcomes among the total patient cohort. Additionally, prediction of cardiovascular death within 2 years was significantly improved by both assays.

Since immunoassays have improved to allow for more sensitive detection of troponin, a considerable proportion of patients without AMI present with troponin levels above the 99th percentile [29]. In studies of patients with acute and chronic heart failure troponin predicted adverse outcomes [30], and even in hospitalized patients without known cardiac disease troponin was independently associated with increased mortality [31]. Recently, the investigators of a large Swedish study of 19,460 patients with chest pain but no myocardial infarction found a strong association of hs-TnT levels with risk for myocardial infarction, heart failure, and mortality [32]. Our findings corroborate these results, demonstrating an independent correlation of troponin I levels with adverse outcomes, particularly in patients without AMI. However, patients with elevated hs-TnI levels but without AMI are seldomly assessed for underlying pathological conditions, e.g., cardiovascular disease, although they are at increased risk for cardiovascular events [33]. At the same time, there is limited knowledge concerning treatment options and explicit investigation for this group of patients [11]. Prospective randomized studies are needed to evaluate the benefit of including troponin measurement into models for cardiovascular risk stratification. The application of enhanced primary prevention in patients with elevated high-sensitivity troponin levels, but without AMI, might constitute a conclusion of our results for clinical practice. However, not only patients with troponin levels above the 99th percentile might benefit from troponin-guided prevention strategies. Previous data suggests that the risk for adverse events increases with troponin levels well below the 99th percentile [12]. In the present analysis, we found a significantly higher overall mortality for patients in the second third of troponin levels compared to the first third, which is surprising given the low median troponin levels in these subgroups (2.3 ng/L for Singulex and 6.8 ng/L for Abbott). The presentation to the emergency department of patients with suspected AMI therefore might be a unique opportunity to plan further diagnostics or preventive therapies to avoid fatal outcomes in these patients. Especially in patients with ruled out AMI, the troponin level, acquired for diagnostic reasons, might be of further use for risk prediction.

In the subgroup of patients with confirmed diagnosis of AMI a single hs-TnI measurement on admission was not predictive for all-cause mortality in the present study, irrespective of the hs-TnI assay used. This finding is in conflict with previous studies that found a considerable prognostic impact of less sensitive troponin assays in patients with AMI [34]. In light of the hs-Tn assays, our findings might have two explanations. First, other than non-AMI patients, patients with AMI usually receive specific treatment in terms of urgent coronary revascularization. Thus, the prognosis of AMI patients is primarily determined by procedural timing and success [35]. Second, it is a persistently high troponin level [36] or a high peak troponin level [37] after AMI, rather than the initial troponin level on admission, that is associated with increased risk for mortality in patients with AMI. Thus, the sole initial hs-TnI level collected from patients with confirmed AMI might not help in stratifying patients at risk for poor overall outcome. However, we found a significant association with troponin I and death of cardiovascular events after two years in AMI patients. Considering the high prevalence of cardiovascular risk factors in AMI patients, the initial troponin level might be a useful criterion to trigger more intensive secondary prevention strategies in AMI patients.

Regarding the prediction of fatal outcomes in addition to classical cardiovascular risk factors, our study found no difference between the established hs-TnI assay and the novel hs-TnI assay, despite the ability of the latter to measure troponin I at even lower concentrations. Since patients with troponin levels below 5 ng/L, analyzed by the Abbott assay, are known to be at very low risk for future cardiovascular events, the additional sensitivity of the Singulex assay might not help in risk prediction. However, in contrast to the Abbott assay, we found a significant difference between patients in the lowest and the middle third of troponin I regarding cardiovascular mortality according to the Singulex assay. Further investigation assessing an additional prognostic value for the measurement of very low troponin concentrations is therefore warranted. 

Our study has several strengths and limitations. By combining two large cohort studies of patients suggestive of AMI, we were able to recruit a relevant sample size and reduce the influence of recruitment bias or local differences in patient management. Although inclusion criteria and protocols of the two studies were highly similar, disagreement between the two studies occurred. Especially, the difference in troponin assays used for adjudication of the final diagnosis might have led to differential adjudication in the two studies. However, differences in patient characteristics were small and we therefore assume they did not affect our results. Our findings are limited exclusively to the described assays and are not transferable to other troponin assays. 

## 6. Conclusions

Troponin I assayed by two high-sensitive assays on admission proved to be an independent predictor for fatal outcome in patients with suspected myocardial infarction. Both assays performed equally in predicting adverse events. 

## 7. Disclosures

Abbott Diagnostics and Singulex provided test reagents for hs-TnI measurements in both studies. J.T.N. received honoraria from Siemens and Abbott Diagnostics, S.B. received honoraria from Abbott Diagnostics, Siemens, Thermo Fisher, and Roche Diagnostics and is a consultant for Thermo Fisher. None of the other authors declared any conflict of interest related to this study.

## Figures and Tables

**Figure 1 biomolecules-09-00469-f001:**
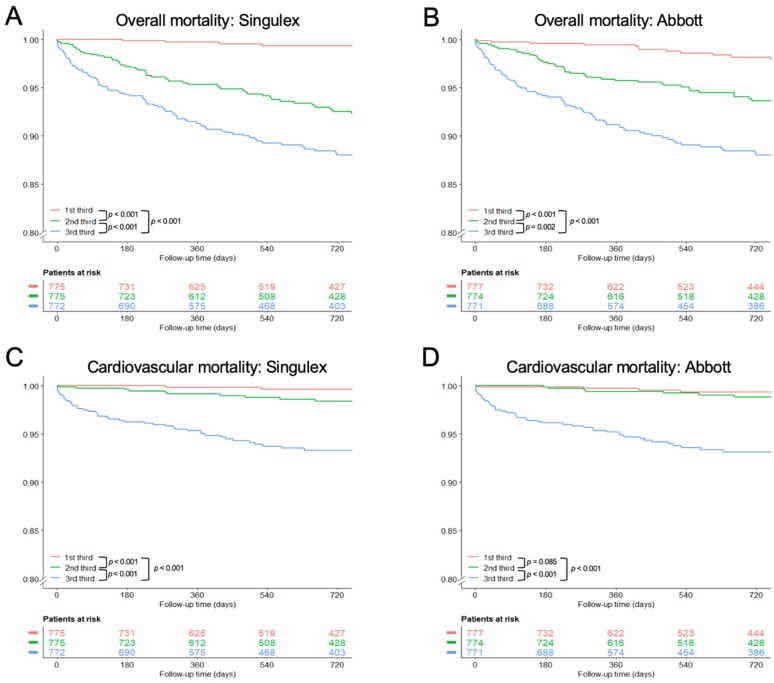
Survival analysis according to troponin thirds. Kaplan–Meier curves according thirds for the endpoints overall (**A**,**B**) and cardiovascular mortality (**C**,**D**). *p* stands for *p*-value of the log-rank test.

**Figure 2 biomolecules-09-00469-f002:**
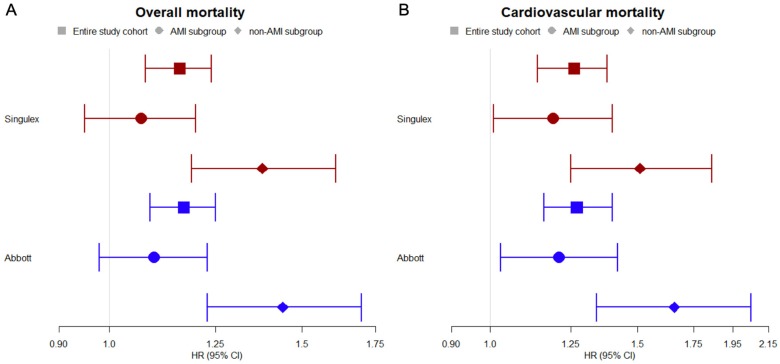
Cox regression analysis for overall mortality (**A**) and cardiovascular mortality (**B**) in the entire study cohort and in subgroups: AMI and non-AMI. Troponin levels were log-transformed. For adjustment variables of sex, age, and conventional cardiovascular risk factors body mass index, diabetes, smoking status, family history of coronary artery disease, hypertension, and hyperlipidemia were used. HR stands for hazard ratio. CI stands for confidence interval.

**Table 1 biomolecules-09-00469-t001:** Baseline characteristics.

	All (*n* = 2332)	AMI (*n* = 518)	non-AMI (*n* = 1814)	*p*-Value
Age (years)	64.0 (52.0, 74.0)	68.0 (58.0, 75.0)	62.0 (50.0, 73.0)	<0.001
Sex (male) No. (%)	1508 (64.7)	368 (71.0)	1140 (62.8)	<0.001
Current smoker No. (%)	567 (24.5)	161 (31.2)	406 (22.5)	<0.001
Diabetes No. (%)	334 (14.5)	95 (18.5)	239 (13.3)	0.0041
Body mass index (kg/m²)	26.5 (24.0, 30.0)	26.7 (24.1, 29.8)	26.4 (23.9, 30.1)	0.71
Hypertension No. (%)	1610 (69.2)	397 (76.8)	1213 (67.0)	<0.001
Dyslipoproteinemia No. (%)	1129 (48.4)	287 (55.4)	842 (46.4)	<0.001
History of AMI No. (%)	412 (17.8)	112 (21.7)	300 (16.6)	0.0093
History of coronary artery disease No. (%)	791 (34.1)	200 (38.6)	591 (32.8)	0.015
Family history of coronary artery disease No. (%)	521 (23.0)	110 (22.0)	411 (23.3)	0.59
Atrial fibrillation No. (%)	382 (16.5)	81 (15.7)	301 (16.7)	0.65
Heart failure No. (%)	217 (9.4)	64 (12.4)	153 (8.5)	0.0088
eGFR (mL/min for 1.73 m²)	77.6 (61.4, 91.8)	70.1 (54.9, 85.5)	80.3 (64.0, 92.9)	<0.001
GRACE Score > 140 No. (%)	245 (10.9)	96 (19.3)	149 (8.5)	<0.001
hs-TnI 0h Singulex (ng/L)	2.3 (1.0, 10.4)	88.3 (13.8, 616.9)	1.6 (0.8, 3.8)	<0.001
hs-TnI 0h Abbott (ng/L)	6.8 (3.1, 25.3)	182.8 (26.7, 1404.6)	5.0 (2.6, 10.6)	<0.001

Baseline characteristics are presented as absolute and relative frequencies for categorical variables and quartiles for continuous variables. AMI stands for acute myocardial infarction. eGFR stands for estimated glomerular filtration rate. GRACE stands for Global Registry of Acute Coronary Events.

**Table 2 biomolecules-09-00469-t002:** C-statistics.

	Prediction Models	C-Index
Overall mortality	Age, sex, CVRFs	0.761 (0.709, 0.813)
	Age, sex, CVRFs + TnI (Singulex)	0.771 (0.719, 0.824)
	C-index difference; *p*-value	0.01; 0.08
	Age, sex, CVRFs	0.761 (0.709, 0.813)
	Age, sex, CVRFs + TnI (Abbott)	0.773 (0.720, 0.825)
	C-index difference; *p*-value	0.012; 0.074
Cardiovascular mortality	Age, sex, CVRFs	0.783 (0.706, 0.860)
	Age, sex, CVRFs + TnI (Singulex)	0.821 (0.744, 0.898)
	C-index difference; *p*-value	0.038; 0.001
	Age, sex, CVRFs	0.783 (0.706, 0.860)
	Age, sex, CVRFs + TnI (Abbott)	0.825 (0.748, 0.902)
	C-index difference; *p*-value	0.042; <0.001

C-statistics for overall mortality and cardiovascular mortality analyzing the additional predictive value of troponin I to a prediction model using age, sex, and cardiovascular risk factors (CVRFs). 95% Confidence intervals are given in parenthesis.

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
