# Peer review of "Prognostic Value of a Novel and Established High-Sensitivity Troponin I Assay in Patients Presenting with Suspected Myocardial Infarction"

_biomolecules, 2019, doi:10.3390/biom9090469_

Round 1
Reviewer 1 Report
The authors present a multicentric study including patients form two prospective cohort studies.
My main concerns about the manuscript are about the study design and methods:
-Inclusion criteria are not exhibited for the StenoCardia Study, so comparison between the two cohorts can not be establihed.
-In-house troponin determination for the diagnosis of AMI are different between the BACC and the StenoCardia study. This limitation should be explained as well as its impact in the diagnostic grouping.
-It is not explained whether patients are divided into thirds acording the troponin level. It suposes that patients from AMI (only 518) and non-AMI groups are forced to be included in the same third, taking into account that median levels form AMI and non-AMI groups are greatly and significantly differents. Please consider quartiles or, even better, use ROC curves to establish a useful cut-off value.
-Attending cox regression it makes no sense to consider body mass index (p:0.71) or family history of coronary artery disease (p: 0.59), since both factors are shown to be non sinificantly different between AMI and non-AMI groups (table 1).
-Please indicate whether prediction of adverse outcome was performed, as it seems, in for entire cohort.
-Figure S1. Please use a cut-off from ROC curves in stead of thirds. A tendency is clear for C and D figures. Perhaps sample size is also relevant in the three subgroups.
-For all Kaplan-Meier curves, please include log-rank comparisons by pairs between the three categories.
Minor aspects:
Please, revise english and grammar. I. e. Table s1. "Baseline characteristics of patients devided..."
"*indicares levels..."
Reviewer 2 Report
What follows is my review of the manuscript entitled “Prognostic value of a novel and an established high sensitivity Troponin I assay in patients presenting with suspected myocardial infarction”, by Dr. Nils A. Sorensen, et al.
In this study, the authors compare a novel with an established high sensitivity Troponin I assay in patients presenting with symptoms suggestive of myocardial infarction. Both troponin measurement determinations were made in 2332 patients who were evaluated at 3 emergency departments in Germany. Higher overall and cardiac mortality were seen on patients with highest Troponin I levels compared with those with lower troponin levels, regardless of which Troponin assay was done.
The methodology of the study is solid, and the number of patients is sufficient to provide meaningful data. I particularly liked the discussion, which gets to the core of the value of a troponin determination, as well as its limitations.
Reviewer 3 Report
The study by Sörensen, Ludwig et al. confirms the present knowledge about prognostic value of high-sensetive Troponin I assay in 2332 patients with suspected acute myocardial infarction. Moreover, they extend the current knowledge by investigating the use of a novel Singulex Clarity cTnI assay. The manuscript is well-written and data analysis is impressive. I believe a few comments still need to be addressed.
Major comments:
1. It would help if the authors could explain a bit better in the introduction section about the similarites and differences between cTnI and cTnT, and in their role in cardiovascular risk prediction.
2. Could you please add correlation analysis between hs-TNI assays (Roche vs. Singulex)? And also report the correltaion between hs-TNI and hs-TNT?
Subgroup analysis for AMI and non-AMI would be of great interest.
3. In order to compare between the two methods it is recommended for you to use the Bland–Altman statistical analysis. (report the standard deviation of the bias and the equasion for deming regression line).
4. Do you have information regarding Major adverse Cardiovascular Events (MACE)? It could provide further support to your findings.
Minor commnets:
1. Please add the recently published NEJM paper (N Engl J Med 2019; 380:2529-2540 DOI: 10.1056/NEJMoa1803377)
2. Make sure to use accronym 'hs-TnI' throught the whole paper.
Round 2
Reviewer 1 Report
The authors response and changes are detailed and well performed and I feel the quality of the manuscript is now good enough to be published in Biomolecules.
Reviewer 3 Report
I do not have further comments.